# Rolling Bearing Fault Diagnosis Using Hybrid Neural Network with Principal Component Analysis

**DOI:** 10.3390/s22228906

**Published:** 2022-11-17

**Authors:** Keshun You, Guangqi Qiu, Yingkui Gu

**Affiliations:** School of Mechanical and Electrical Engineering, Jiangxi University of Science and Technology, Ganzhou 341000, China

**Keywords:** PHM, intelligent fault diagnosis, complex extreme variable loading, hybrid deep neural network, robustness and generality

## Abstract

With the rapid development of fault prognostics and health management (PHM) technology, more and more deep learning algorithms have been applied to the intelligent fault diagnosis of rolling bearings, and although all of them can achieve over 90% diagnostic accuracy, the generality and robustness of the models cannot be truly verified under complex extreme variable loading conditions. In this study, an end-to-end rolling bearing fault diagnosis model of a hybrid deep neural network with principal component analysis is proposed. Firstly, in order to reduce the complexity of deep learning computation, data pre-processing is performed by principal component analysis (PCA) with feature dimensionality reduction. The preprocessed data is imported into the hybrid deep learning model. The first layer of the model uses a CNN algorithm for denoising and simple feature extraction, the second layer makes use of bi-directional long and short memory (BiLSTM) for greater in-depth extraction of the data with time series features, and the last layer uses an attention mechanism for optimal weight assignment, which can further improve the diagnostic precision. The test accuracy of this model is fully comparable to existing deep learning fault diagnosis models, especially under low load; the test accuracy is 100% at constant load and nearly 90% for variable load, and the test accuracy is 72.8% at extreme variable load (2.205 N·m/s–0.735 N·m/s and 0.735 N·m/s–2.205 N·m/s), which are the worst possible load conditions. The experimental results fully prove that the model has reliable robustness and generality.

## 1. Introduction

Due to complex working conditions and frequently changing loads in actual production, a large number of mechanical system failures are caused by faults in bearings [1]. The mechanism of bearing damage is very complex; the machine operating environment [2], frequent fluctuations in load [3,4,5], and improper installation, etc., can all cause different types of bearing faults, mainly including abrasion failure, fatigue failure, corrosion failure, and cavitation failure [6]. It is very difficult and unrealistic to analyze and diagnose faults by only studying the mechanism [7], but some studies have modeled bearing dynamics in terms of the radial internal clearance of rolling bearings as a way of analyzing bearing failure and life [8,9], which provide good references. Therefore, we can combine mechanism analyses to research a better intelligent fault diagnosis method. Rolling bearings, as important rotating parts in machinery and equipment, are also one of the important sources of faults in machinery and equipment [10]. Rolling bearings are one of the most common and widely used kinds of bearing; therefore, the fault diagnosis method of rolling bearings has been one of the key technologies in the development of machinery fault diagnosis [11].

Fault prognostic and health management (PHM) systems need to have a complete, practical, intelligent, reliable, and systematic solution for rolling bearing health management [12,13], which includes raw data pre-processing, feature value selection and extraction, failure mode identification, performance degradation assessment, performance trend prediction, and maintenance decision making [14]. Raw data pre-processing mainly includes outlier processing, zero-meaning processing, trend term elimination, and digital filtering [15]. Eigenvalue selection and extraction mainly includes time domain analysis, frequency domain analysis, time-frequency domain analysis, data constraint methods, and sensor information fusion [16,17]. Fault diagnosis mainly includes knowledge-based expert systems, analytic model-based state estimation, parameter estimation, and equivalence space [18], as well as data-driven neural networks, support vector machines (SVM) [19], information fusion, and multivariate statistical analysis [20]. Fault prediction mainly includes traditional reliability prediction methods based on fault tree analysis and lifetime distribution models, failure mechanism models based on physical methods of failure (FMM), Bayesian models based on statistics, Hidden Markov Models (HMM), and data-driven autoregressive models, neural network regression models, and support vector regression (SVR) models [21]. Performance trends are mainly predicted and evaluated in terms of model and algorithm errors. 

In this paper, we propose a data-driven hybrid neural-network-based intelligent multi-classification algorithm that can automatically extract and process features, learn, reason, and decide whether a rolling bearing is normal or faulty, as well as what kind of fault it is, directly from the raw vibration data. The PCA method is used as data pre-processing to reduce computational complexity and improve feature extraction, and a convolutional neural network (CNN) algorithm with strong feature extraction capability is designed in the first layer of the network, which can autonomously design the size of the convolutional layers and enable initial learning and inference of the data. Then, a bi-directional Long Short Time Memory (BiLSTM) structure is designed in the second layer, which maximizes the learning performance of the model in learning time series data features. In order to reduce the number of parameters computed by the model, and to improve the speed and diagnostic performance of the model, an attention-based soft classification algorithm is designed in the last layer. In order to test the generality of the proposed model, we trained the model using operating data at 1 horsepower (1 hp = 0.735 N·m/s) load and operating conditions, conducted experimental tests using operating data at 0–2.205 N·m/s, and finally, we used bearing vibration data at six different variable load conditions (1-2 hp, 1-3 hp, 2-1 hp, 2-3 hp, 3-1 hp, 3-2 hp) for prognostic diagnosis. 

The remainder of the paper is organized as follows: Section 2 analyzes related work involved in the rolling bearing fault diagnosis. Section 3 describes the theory and methods of rolling bearing fault diagnosis through the proposed hybrid deep neural network classification algorithm. Section 4 then analyses the experimental data and results, including a performance comparison with alternative methodologies. Finally, Section 5 concludes key results of the research.

## 2. Related Works 

About 30–40% of equipment failures are caused by bearing failures. Since the 1990s [22], the research of algorithms for fault diagnosis of bearings has been a key topic. From traditional statistical methods to machine learning methods, and recently the application of deep learning algorithms, algorithms have been and continue to be areas of continued improvement and research. 

Before the deep learning boom, classical machine learning (ML)-based algorithms were sensibly used in industrial bearing fault diagnosis [21]. For example: artificial neural networks (ANN) have been applied for more than 30 years [23]; principal component analysis (PCA) also has a great advantage in the extraction of sensitive features of bearing fault data [24,25]; K-Nearest Neighbors (K-NN) has been applied in bearing fault diagnosis for more than a decade [26]; and SVM has been famous for its powerful nonlinear processing capability with good generalization performance since its introduction. SVM is renowned for its powerful nonlinear processing ability, good generalization performance, being suitable for learning small sample data, and performance in bearing fault diagnosis, which is far better than ANN, KNN, etc. [27], and it is often used in combination with other algorithms in practical research to achieve better diagnosis results [28]. Furthermore, neuro-fuzzy networks [29], Bayesian networks [16], self-organizing maps [30], extreme learning machines (ELM) [31], transfer learning [32], linear discriminant analysis [33], quadratic discriminant analysis [34], random forests [35], independent component analysis [36], empirical pattern decomposition [37], correlation analysis [38], affinity propagation [39], and dictionary learning [40] have all been widely used in bearing fault diagnosis, among others.

With the advent of big data and the development of high performance processors, deep learning algorithms have emerged in bearing fault diagnosis. The CNN’s excellent denoising and classification performance, which has been widely used in image processing, has also been gradually applied to bearing fault diagnosis [41]. Examples include Adaptive CNN (ADCNN) [42], LeNet-5 based CNN [43], Deep Fully Convolutional Neural Network (DFCNN) [44], Multiscale CNN (MS-DCNN) [45], Pythagorean Spatial Pyramidal Pooling (PSPP) CNN [46], and Adaptive Overlapping CNN (AOCNN) [47]. With the demand for unsupervised learning, a combination of auto-encoders (AE) were developed for application to bearing fault diagnosis [48]. Various deep learning convolutional neural networks have been developed and applied to fault diagnosis, such as the Deep Belief Network (DBN) [49], the Recurrent Neural Network (RNN) [50], the Generative Adversarial Network (GAN) [51], Long Short Time Memory (LSTM) [52], transfer learning based deep learning [53], the Graph Neural Network (GNN) [54], and their variants, etc. 

The accuracy of bearing fault diagnosis based on advanced deep learning algorithms has reached excellent accuracy of over 90%, but more efficient, versatile, and more precisely accurate hybrid deep learning neural network models are needed to cope with fault diagnosis problems under various variable load conditions.

## 3. Introduction to Theory and Methodology

### 3.1. Level I: Fault Generation of Rolling Bearings

Most bearings cannot reach the designed life during operation, mainly because of poor lubrication, unreasonable assembly, and manufacturing defects. In order to diagnose the failure of the bearings during operation, we generally use more advanced sensors to obtain the vibration signal of the corresponding position, and then combine the signal processing method.

As shown in Figure 1, first assume that the number of bearing balls is *Z*, the diameter of balls is *d*, the bearing raceway pitch is *D*, the bearing contact angle is α, the inner raceway radius is *r_1_*, the outer raceway radius is *r_2_*, and the bearing inner ring speed is *n*. Theoretically, the characteristic frequency equation of rolling bearing has the following:

Inner ring rotation frequency *f_i_*
(1)$$f_{i}=\frac{n}{60}$$

Relative rotation frequency $\;f_{r}$ of inner and outer ring, because the outer ring of the rolling bearing does not rotate, the outer ring rotation frequency $\;f_{0}\;$ is 0,
(2)$$\;f_{r}=f_{i}-f_{0}=f_{i}$$

Frequency $f_{ic}$ of rolling body passing a point of inner ring,
(3)$$\;\;\;\;f_{ic}=\frac{1}{2}Z\left(1+\frac{d}{D}\cos\alpha \right)f_{r}$$

Frequency $f_{oc}$ of rolling body passing a point on the outer ring,
(4)$$\;\;\;\;f_{b}=\frac{1}{2}Z\left(1-\frac{d}{D}\cos\alpha \right)f_{r}$$

Rotational frequency $\;f_{b}$ of the rolling body. The calculation formula is equivalent to the calculation formula of cage rotation frequency $f_{c}$.
(5)$$\;\;\;\;f_{b}=\frac{1}{2}\left(1-\frac{d}{D}\cos\alpha \right)f_{r}$$

When a fault occurs, the fault frequency of the bearing can be empirically calculated, at which time the fault frequency of the inner ring becomes:(6)$$\;\;\;\;\;f_{i}=0.6\times Z\times f_{r}$$

Fault frequency of outer ring:(7)$$\;\;\;\;\;f_{0}=0.4\times Z\times f_{r}$$

Frequency of cage faults:(8)$$\;\;\;\;\;f_{c}=0.381-0.4\times f_{r}$$

Frequency of rolling body faults:(9)$$f_{c}=\left\{\begin{array}{c} 0.23\times Z\times f_{r},\;\;\;\;\;\;Z<0\\ 0.18\times Z\times f_{r},\;\;\;\;\;\;Z\geq 10 \end{array}\right.$$

Frequency relationship between outer ring and cage:(10)$$\;\;\;\;\;f_{0}=Z\times f_{c}$$

Frequency relationship between outer ring and inner ring:(11)$$f_{0}+f_{i}=Z\times f_{r}$$

Based on the analysis of the failure mechanism of rolling bearings, combined with the actual production experience derived from the failure formula, to a certain extent, we can indeed produce a practical guidance, but the role produced is limited; the biggest drawback is it is unreliable and precision diagnosis is too low.

### 3.2. Level II: Fault Diagnosis Methods

Bearing fault diagnosis has been a popular area of research, and an algorithm usually includes two parts: signal feature extraction and classification. Common feature extraction algorithms include fast Fourier variation, wavelet transform, empirical pattern decomposition, and statistical features of the signal, etc. As shown in Figure 2, the intelligent diagnosis method based on machine signal processing through feature extraction algorithm combined with classifier requires expert experience, a time-consuming design, and cannot guarantee generality, and thus, it is difficult to meet the requirements of large data and accuracy. 

### 3.3. Level III: The Proposed Hybrid Neural Network Fault Diagnosis Method with PCA

#### 3.3.1. Data Pre-Processing with PCA

PCA is the most commonly used linear dimensionality reduction method. The goal is to map high-dimensional data into a low-dimensional space by linear projection and expect the maximum amount of information (maximum variance) in the projected dimension, so as to use fewer data dimensions while retaining the characteristics of more original data points. The purpose is to reduce the noise or computational effort of the data while trying to ensure that the amount of information is not distorted.

First, assume that the data set *X* = [*x_1_*, *x_2_*,…, *x_n_*] has *n* sets of data and each set has *m* features.
(1)Normalization of the data, i.e.,


(12)
$$Z_{ij}=\frac{x_{ij}-{\bar{x}}_{j}}{\sqrt{\sigma \left(x_{j}\right)}}$$


(2)Calculating the covariance matrix *C* of the normalized data.


(13)
$$C\left(X_{i},X_{j}\right)=\frac{\sum_{k=1}^{n}\left(x_{i}^{k}-{\bar{x}}_{i}^{k}\right)(x_{j}^{k}-{\bar{x}}_{j}^{k})}{m-1}$$


(3)Computing the eigenvalues (*λ_1_*, *λ_2_*,…, *λ_m_*) of the covariance matrix *C*.


(14)
Cu=λu


(4)Calculating the cumulative contribution of the first *k* principal components. When the cumulative contribution rate $Ø\left(p\right)$ ≥ 90%, only the first *k* feature vectors can be extracted as sample features, and the larger the cumulative contribution rate is, the more original information is included.


(15)
$$Ø\left(p\right)=\sum_{i=1}^{p}\lambda_{i}/\sum_{i=1}^{m}\lambda_{i}$$


(5)Deriving the principal component feature vector *Y* with reduced dimensional.


(16)
$$Y=\left[\begin{array}{l} y_{i}^{i}\\ \;•\\ \;•\\ \;•\\ y_{k}^{i} \end{array}\right]=\left[\begin{array}{c} u_{1}^{T}.{\left(x_{1}^{i},x_{2}^{i},\ldots ,x_{n}^{i}\right)}^{T}\\ u_{2}^{T}.{\left(x_{1}^{i},x_{2}^{i},\ldots ,x_{n}^{i}\right)}^{T}\\ •\\ •\\ •\\ u_{k}^{T}.{\left(x_{1}^{i},x_{2}^{i},\ldots ,x_{n}^{i}\right)}^{T} \end{array}\right]$$


As shown in Figure 3, the original feature data is 470 dimensions, and the 330 most sensitive features can be obtained by calculating the cumulative contribution rate of 90%. By using this data pre-processing method with PCA, we can achieve a successful feature dimension reduction of 30%, which will effectively reduce the workload of the later deep learning computation and will also produce some denoising. 

#### 3.3.2. End-to-End Hybrid Neural Network Fault Diagnosis Method

The proposed end-to-end hybrid neural-network-based intelligent diagnosis algorithm is used to simultaneously complete feature extraction and fault detection, which integrates all the advantages of CNN, LSTM, and attention mechanisms. Being a hybrid deep learning convolutional neural network model, it has already achieved good prospects in medical biology and other applications [55], and still continues to prosper in engineering applications.

As shown in Figure 4, the first layer of the model has a 1-dimensional CNN convolutional layer, which is mainly responsible for the denoising and feature extraction of the original vibration data; its calculation is shown in Equation (17).
(17)$$\left(i,j\right)=I*K\left(i,j\right)=\underset{m}{\sum }\underset{n}{\sum }I\left(i+m,j+n\right)K\left(m,n\right)$$
where *S* represents the result of the operation; *I* is the original image; *K* is the convolution kernel; *m*, *n* are the height and width of the convolution kernel; *i*, *j* represent the position of the convolved.

The second layer is designed as a bi-directional LSTM (Bi-LSTM), which is a type of recurrent neural network (RNN). In practice, RNNs have been found to have problems such as gradient disappearance, gradient explosion, and a poor ability to rely on information over long distances; thus, the LSTM was introduced. The LSTM is similar to a RNN in terms of its main structure, but the main improvement is the addition of three gates in the hidden layer *h*, which are a forgetting gate, input gate, and output gate, as well as the addition of a new cell state. The principle is shown in Figure 4. $f\left(t\right)$, $i\left(t\right)$, and *o*(*t*) represent the values of the forgetting gate, input gate, and output gate at time *t*, respectively. $\alpha \left(t\right)$ denotes the initial feature extraction of *h*(*t* − 1) and *x*(*t*) at time *t*. The specific calculation process is shown in Equations (18)–(21).
(18)$$f\left(t\right)=\sigma \left(Wfh_{t-1}+Ufx_{t}+bf\right)$$
(19)$$i\left(t\right)=\sigma \left(W_{i}h_{t-1}+U_{i}x_{t}+b_{i}\right)$$
(20)$$\alpha \left(t\right)=\tan h\left(W_{a}h_{t-1}+U_{a}x_{t}+b_{a}\right)$$
(21)$$o\left(t\right)=\sigma \left(W_{0}h_{t-1}+U_{0}x_{t}+b_{0}\right)$$
where *x_t_* denotes the input at time *t*; *h_t_*_ − 1_ denotes the hidden state value at time *t* − 1; *Wf*, *Wi*, *Wo*, and *Wa* denote the weight coefficients of *h_t_*_–1_ in the forgetting gate, input gate, output gate, and feature extraction process, respectively; *U_f_*, *U_i_*, *U_o_*, and *U_a_* denote the weight coefficients of *x_t_* in the forgetting gate, input gate, output gate, and feature extraction process, respectively; *b_f_*, *b_i_*, *b_o_*, and *b_a_* denote the bias values of *x_t_* in the forgetting gate, input gate, output gate, and feature extraction process, respectively; *U_f_*, *U_i_*, *U_o_*, and *U_a_* denote the weight coefficients of the forgetting gate, input gate, output gate, and feature extraction process *x_t_*, respectively; *b_f_*, *b_i_*, *b_o_*, and *b_a_* denote the bias values of the forgetting gate, input gate, output gate, and feature extraction process, respectively; tan*h* denotes the tangent hyperbolic function and *σ* denotes the activation function Sigmoid.
(22)$$\tan h\left(x\right)=\frac{1-e^{-2x}}{1+e^{-2x}}$$
(23)$$\sigma \left(x\right)=\frac{1}{1+e^{-x}}$$

The result of the forgetting gate and input gate calculation acting on *c*(*t* − 1) constitutes the cell state *c*(*t*) at time *t*, which is expressed in Equation (19) as:(24)$$c\left(t\right)=c\left(t-1\right)⊙f\left(t\right)+i\left(t\right)⊙\alpha \left(t\right)$$
where ⊙ is the Hadamard product. Eventually, the hidden layer state *h*(*t*) at time t is solved by the output gate *o*(*t*) and the cell state *c*(*t*) at the current moment.
(25)$$h\left(t\right)=o\left(t\right)⊙\tan h\left(c(t\right))$$

As shown in Figure 5, the BiLSTM neural network structure model consists of two independent LSTM, as shown in Figure 6. The input sequences are input into the two LSTM in positive and negative order, respectively, for feature extraction, and the two output vectors (i.e., the extracted feature vectors) are stitched together to form the final feature expression of the output. 

The BiLSTM model is designed so that the feature data obtained at time *t* has information between the past and the future at the same time. Experimentally, this neural network structure has proven to be more efficient and perform better than a single LSTM structure for feature extraction of time series data. It is worth mentioning that the parameters of the 2 LSTM neural networks in BiLSTM are independent of each other.

As shown in Figure 7, when the feature data extracted by BiLSTM is fed to the Attention mechanism layer, the Attention technique causes the data to be classified as more feature-specific by weighting the data with different features and reassigning the weights through the learning and scoring results. Here, the Score is first defined as Equation (26).
(26)$$Score\left(h_{t},\;{\bar{h}}_{s}\right)=h_{t}^{T}\cdot W\cdot \;{\bar{h}}_{s}$$
where $h_{t}$ is the hidden state of the decoder at time *t*, and ${\bar{h}}_{s}$ denotes the hidden states of the encoder, *W* is a matrix to be learned, which is used throughout the process. After the score is obtained, we can find the weight of attention $\alpha_{ts}$.
(27)$$\alpha_{ts}=\frac{\exp\left(Score\left(h_{t},\;{\bar{h}}_{s}\right)\right)}{\sum_{s^{\prime }=1}^{S}\exp\left(Score\left(h_{t},\;{\bar{h}}_{s’}\right)\right)}$$

Then, the weights are multiplied with the hidden states in the encoder to obtain the feature vector *c_t_*.
(28)$$c_{t}=\underset{s}{\sum }\alpha_{ts}{\bar{h}}_{s}$$

After that, we can calculate the Attention vector $\alpha_{t}$, combined with the weights of attention $\alpha_{ts}$, and the final value of attention can be derived.
(29)$$\alpha_{t}=f\left(c_{t},h_{t}\right)=\tan\left(W_{c}\left[c_{t},h_{t}\right]\right)$$

The Attention Mechanism is an information filtering method that further alleviates the problem of long-term dependency in LSTM and GRU [56]. In general, this can be achieved in three steps: first, a task-relevant representation vector is introduced as a benchmark for feature selection, a manually specified hyperparameter, which can be either a dynamically generated vector or a learnable parameter vector; then, a scoring function is chosen to calculate the correlation between the input features and this vector to obtain the probability distribution of the features being selected, which is called the attention distribution; finally, a weighted average of the input features by the attention distribution filters out the task-relevant feature information.

## 4. Bearing Diagnostic Performance Verification of the Proposed Model

As shown in Figure 8, the experimental platform consists of a drive motor, a torque transducer, and a power tester (right side of the figure) [57]. 

### 4.1. Level I: Introduction to the Conditions and Data Set of the Experiment

Rolling bearing fault diagnosis is generally performed using the CWRU dataset to standardize the strengths and weaknesses of detection algorithms. As shown in Table 1, the data in this experiment uses DE (drive end) accelerometer data and a bearing with SKF6205 type load for 0-3 horsepower (0–2.205 N·m/s) corresponding to the approximate motor speed of 1797 r/min, 1772 r/min, 1750 r/min, and 1730 r/min; the sampling frequency is 48 kHz, the experimental single point damage diameter of the selected bearing is 0.007 mm, 0.014 mm, and 0.021 mm, and each fault diameter contains a rolling body fault, inner ring fault, and outer ring fault [58].

As shown in Table 2, the experimental dataset consists of nine fault datasets and one normal dataset, and the datasets used for training are generated by combining the ten classes of datasets corresponding to 0-3 hp, respectively. The sample size of each class is 256 and the sample size of the combined dataset is 1024; 70% of the combined dataset is used as the training set and 30% as the test set.

As shown in Figure 9, by observing the number of features, magnitude, fluctuation period, and phase difference in the bearing vibration signal, it can be found that the normal bearing vibration is more regular and the period is more stable, but after carefully observing the rolling bearing vibration signal with a fault, it is found that it is difficult to classify the bearing with a fault by manual observation of these data due to the influence of noise, different working conditions, and limited human perception ability. As shown in Figure 10, the linear FFT analysis allows for a rough determination of the frequency that produces the maximum vibration signal; i.e., the most likely frequency of the fault. Although the FFT analysis method gives us a simple and reliable way to diagnose faults directly to the senses, it is limited by problems such as noise in the vibration signal and imbalance in the data, so it is not really a very objective diagnostic method.

### 4.2. Level II: Training Results of the Model and Testing under Different Load Cases

The model is trained using the created dataset, and the trained model is also saved. The accuracy curve and loss rate curve of the model in the training process are shown in Figure 11. We can see that the model is well trained and there is no overfitting phenomenon. This is because, in the training process, we use a 10-fold cross-validation method, which groups the raw dataset into a training set and a validation set or test set. Firstly, divide the dataset into ten parts. Then, take turns to allocate nine of them for training and one for validation, and finally, use the mean of the ten results as an estimate of the accuracy of the algorithm.

To refine the fault diagnosis of the model, the model was tested using data under 0 hp, 1 hp, 2 hp, and 3 hp loads, and a confusion matrix was used to represent the fault diagnosis results. As can be seen from the confusion matrix in Figure 12, only some of the samples are incorrectly identified, and most of them are 100% identified. Combining the four plots, the diagnostic results are a little better under loads of 0 hp, 1 hp, and 2 hp conditions. To more clearly represent the feature extraction ability of the model, t-SNE is introduced to downscale and visualize the features of each network layer of the model; only the feature extraction results of each network layer under the load of 1 hp condition are shown in this paper.

As shown in Figure 13, which provides the t-SNE visualization results of each layer of the model when the input layer is a time-domain signal, the data of the bearings in different operation states are mixed with each other, and their clustering effect is extremely poor. From the t-SNE visualization results of the convolutional layer, we can see that some samples with the same type have already started to aggregate, and, as the network layer goes deeper, the t-SNE visualization results of the BiLSTM of second layer have basically completed the accurate classification of the vast majority of samples, and only a small number of samples are misclassified. Finally, the clustering effect is more obvious in the attention layer of the third layer, which is consistent with the results of the confusion matrix above, and also proves that the model has superior diagnostic ability. As shown in Table 3, the end-to-end rolling bearing fault diagnosis model, based on the hybrid deep neural network proposed in this paper, is not only efficient but also has high accuracy and some advantages compared with other deep learning neural network fault diagnosis methods.

As shown in Figure 14, in order to compare the classification effectiveness of the proposed model with other intelligent fault diagnosis models, four existing, more advanced deep learning fault diagnosis models (SAE, CNN-LSTM, PSPP-CNN, and LeNet-5-CNN) were statistically analyzed. The CWRU dataset was still used for testing, and the objective of the test was to de-classify ten types of rolling bearing faults from 0–9 categories at a motor load of 2.205 N·m/s. The classification results from the t-SNE visualization statistics of the four models show that although they are good enough for fault diagnosis, there is still some gap in classification effectiveness with the hybrid deep learning model proposed in this study.

As shown in Figure 15, to test the generalization performance of the model proposed in this study, this paper conducted a cross-dataset test, using the same amount of data from the real dataset of an official industrial big data competition to test the model. It was also found that the deep learning intelligent fault diagnosis model proposed in this study performs very well in other datasets, basically classifying all types of faults. 

### 4.3. Level III: Diagnostic Performance Verification with Load Variation by Practical Testing

Changes in work load are common for a mechanical system, and when the load changes, the signal measured by the sensor will also change. Under different loads, the number of features in the vibration signal is not the same, and so, the amplitude size is not the same, and the fluctuation period and phase difference are also large. The above situation will cause the classifier to be unable to accurately classify the extracted features, thus reducing the generalization performance of the intelligent fault diagnosis system. In order to verify the diagnostic performance of the model under the changed actual working environment, we built a bearing diagnostic signal acquisition platform, as shown in Figure 16, and put the vibration signal sensor to the DE side for data acquisition. Adjusting the motor speed to 1-3 hp, we recorded the signal with and without the faulty bearing, respectively.

Although the model achieved good diagnostic results under the condition of constant load, in practice, the load of the bearing was variable. To further verify the generalization ability of the model proposed in this paper, the diagnostic performance under load variation conditions were tested in the practical platform. Training samples with loads of 1 hp, 2 hp, and 3 hp were used to train the model, and the remaining test samples were used to test the generalization ability of the model.

The test results are shown in Figure 17. The model has the highest fault diagnosis accuracy under variable loads of 1-2 hp and 2-1 hp, with an average accuracy close to 90%, and the worst fault diagnosis accuracy under variable loads of 3-1 hp, 2-3 hp, with an average accuracy of about 72.8%. On the whole, although the diagnostic accuracy of the model is poor under the variable operating conditions of high load, the diagnostic accuracy is high under the variable operating conditions of low load, and the overall average accuracy is more than 80%. Therefore, it can be seen that the model can be applied to fault diagnosis under conventional load variation conditions.

In this study, to test the nonlinear robustness and generalization capability of the fault classification model, we built a practical test platform and conducted six sets of destructive experiments with variable loads. In actual machine operation, the motor load does not generally drastically fluctuate, so that a generally acceptable bearing will not produce the faults that we want in experiments over a short period of time, even under normal conditions of variable workloads. Since it is to difficult attain a clear indication in the experiment whether a fault has been generated, or what the specific type of fault is, etc., we do not know when to use the sensor to obtain data on the vibration signal. Therefore, the use of good diagnostic model measurements in practice has a certain level of randomness, which makes the actual experimental testing very challenging. Moreover, extreme variable load conditions are difficult to generate in general, and, even under extreme variable load conditions, machine failure does not necessarily immediately occur because the machine will have a certain load carrying capacity.

Through practical experimental tests, we clearly understand that fault diagnosis research under extreme variable load conditions in the study is both significant and a challenge. Therefore, we should consider some randomness issues of the classification model and the practical tests, so as to improve the real performance of the model.

## 5. Conclusions

With the continuous development of fault prognostics and health management (PHM) technology, intelligent fault diagnosis models have emerged. Most of the rolling bearing intelligent diagnosis models based on deep learning have considerable testing accuracy, but there are still problems in terms of insufficient generality and robustness, especially under the working conditions of rapidly changing loads. 

In this study, we first explored the generation of rolling bearing faults and some empirical frequency formulas for bearing fault vibration, and concluded the inefficiency and unreliability of traditional fault diagnosis methods. Therefore, an end-to-end hybrid deep neural-network-based model with PCA was constructed and applied to the fault diagnosis of rolling bearings. When the original unclustered data were input to the model, the data were first pre-processed by feature dimensionality reduction, and then features were extracted and denoised by a CNN algorithm in the first layer, followed by a bi-directional LSTM algorithm for feature extraction and memory of time series data. Finally, the attention mechanism was used to improve the weight of different categories of feature data for Softmax classification to improve the accuracy of diagnosis. 

The CWRU dataset was used to train and test the model, and the experimental results showed that the highest test accuracy of the model under low load conditions was close to 100%; the overall test accuracy was 99.98%, which surpassed most existing deep learning fault diagnosis models. In order to test the diagnostic accuracy of the model under variable load conditions, six sets of achievable experiments were designed. The experimental results showed that this model still had 72.8% diagnostic accuracy under extreme variable load conditions, and more than 80% diagnostic accuracy under overall variable load conditions, which indicated that the diagnostic model has considerable robustness and versatility.

In terms of next steps, we will design more practical test experiments and sufficiently overcome the randomness caused by model testing to conduct repeated cross-platform tests. Moreover, we will do our best to deploy and improve the intelligent fault diagnosis model proposed in this study.

## Figures and Tables

**Figure 1 sensors-22-08906-f001:**
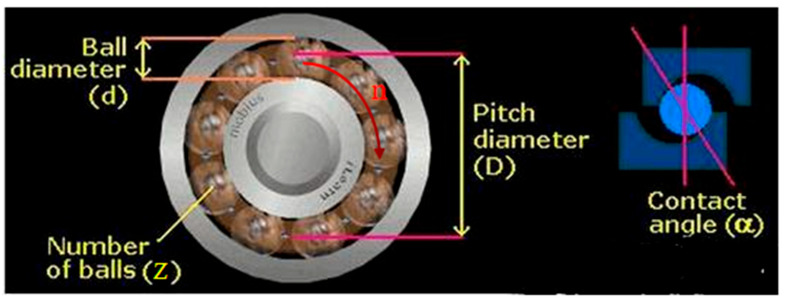
Schematic diagram of the geometric parameters of rolling bearings.

**Figure 2 sensors-22-08906-f002:**
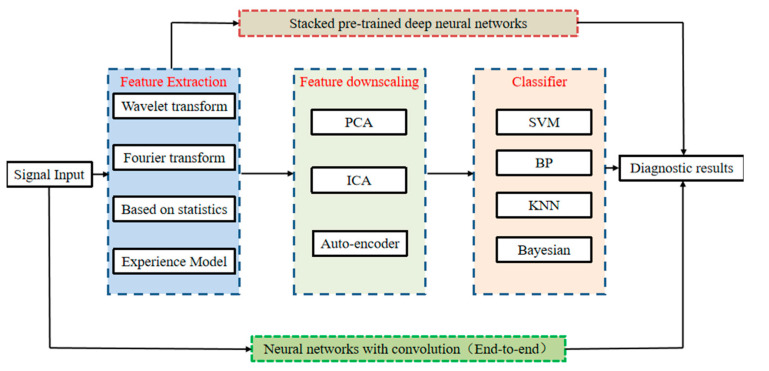
Intelligent fault diagnosis method.

**Figure 3 sensors-22-08906-f003:**
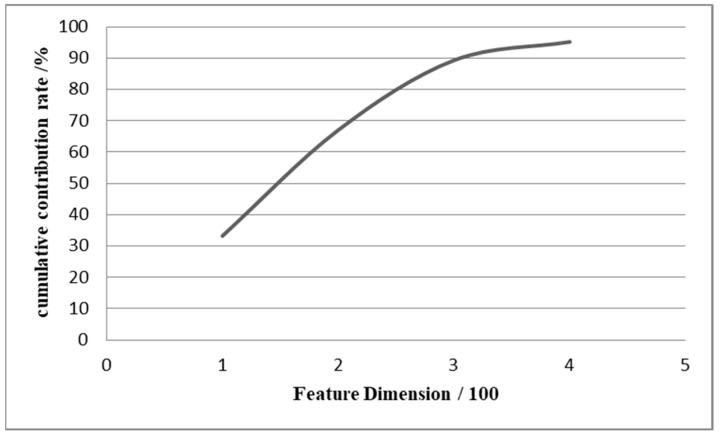
The feature dimension of 90% cumulative contribution rate.

**Figure 4 sensors-22-08906-f004:**
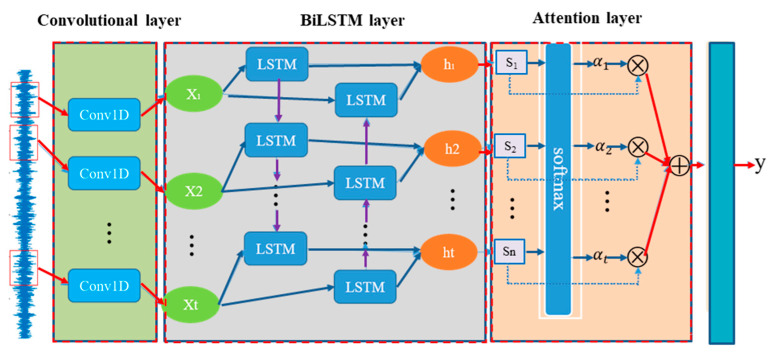
The proposed hybrid neural network fault detection method.

**Figure 5 sensors-22-08906-f005:**
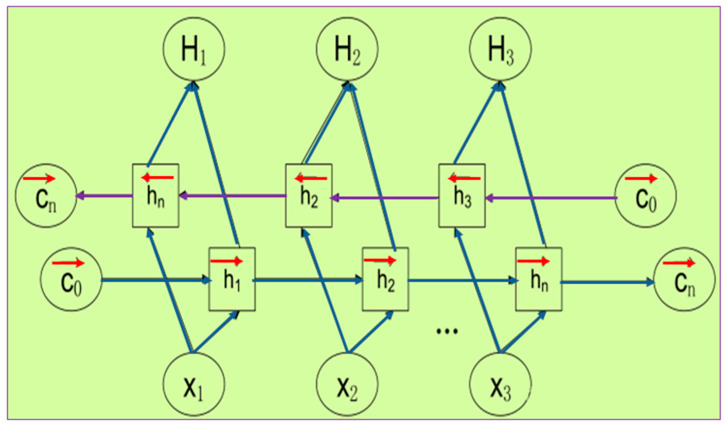
BiLSTM principle.

**Figure 6 sensors-22-08906-f006:**
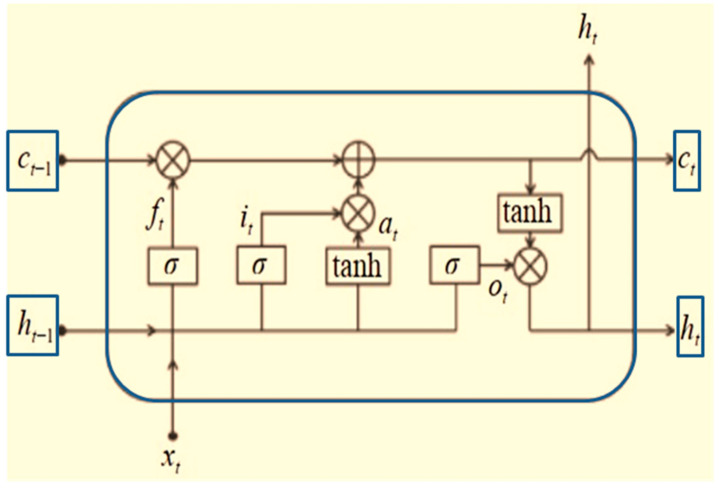
LSTM principle.

**Figure 7 sensors-22-08906-f007:**
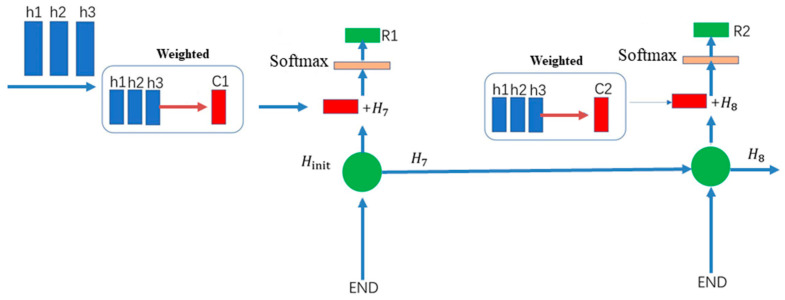
Attention technique.

**Figure 8 sensors-22-08906-f008:**
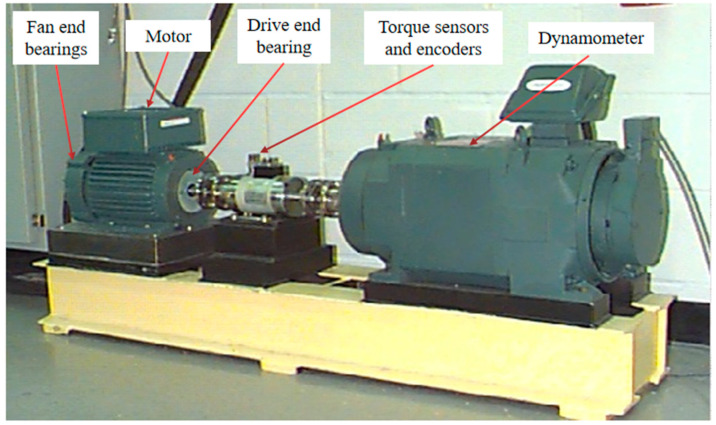
Equipment for experimental data acquisition (from Case Western Reserve University (CWRU)).

**Figure 9 sensors-22-08906-f009:**
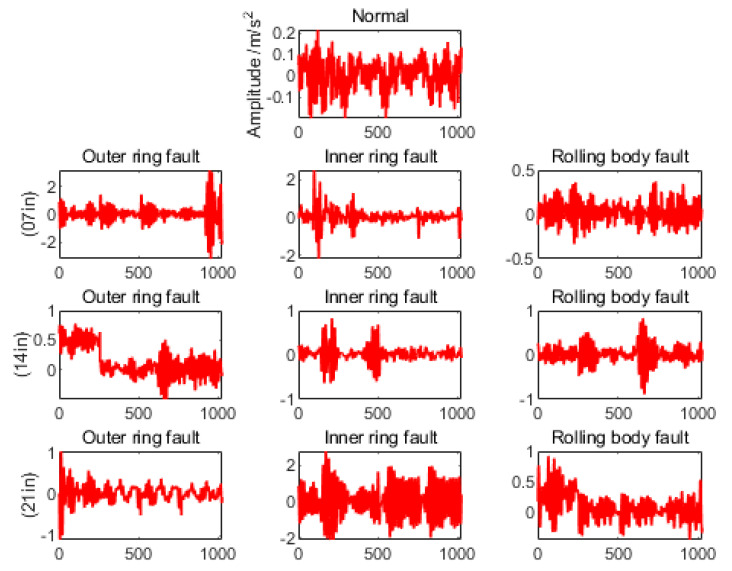
Visual analysis of rolling bearing data (occurring on outer ring, inner ring, and rolling body with single point diameter damage of 0.007 mm (07 in), 0.014 mm (14 in), and 0.021 mm (21 in), respectively).

**Figure 10 sensors-22-08906-f010:**
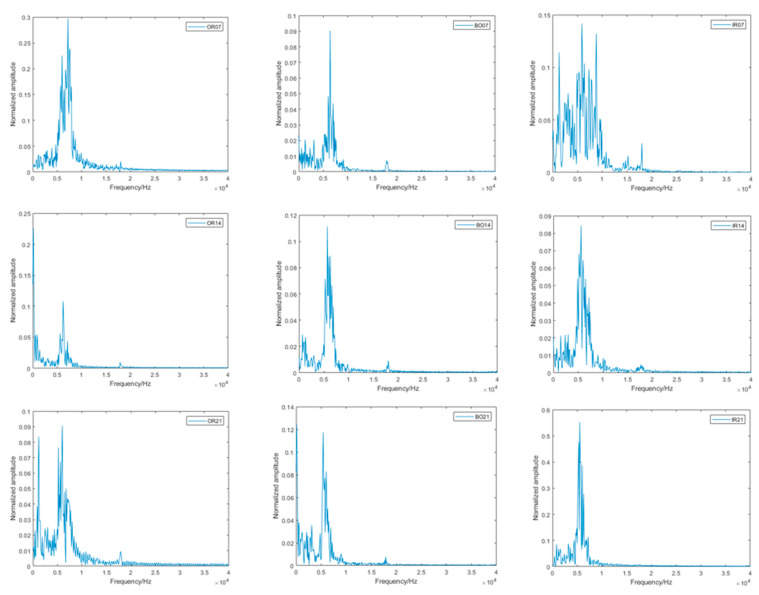
FFT analysis of rolling bearing faults data.

**Figure 11 sensors-22-08906-f011:**
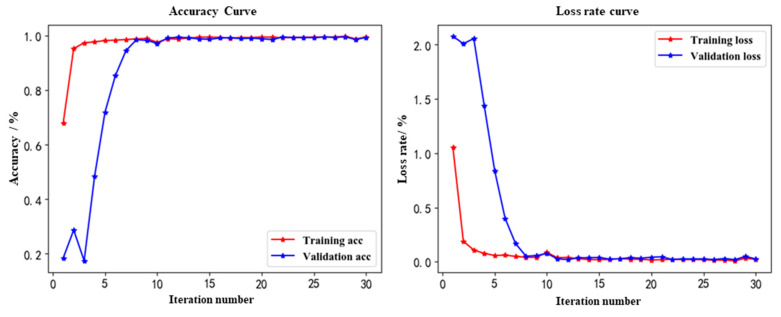
Training result.

**Figure 12 sensors-22-08906-f012:**
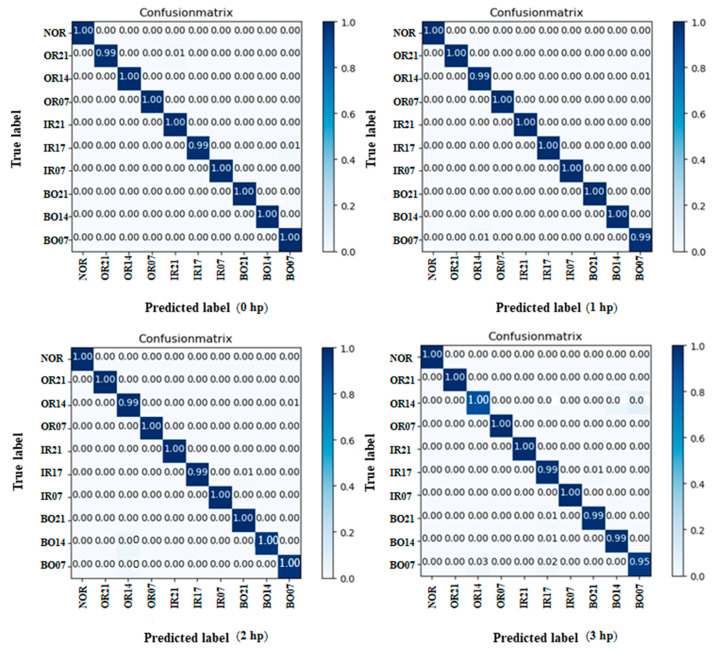
Test results of model under a load of 0.735 N·m/s~2.205 N·m/s.

**Figure 13 sensors-22-08906-f013:**
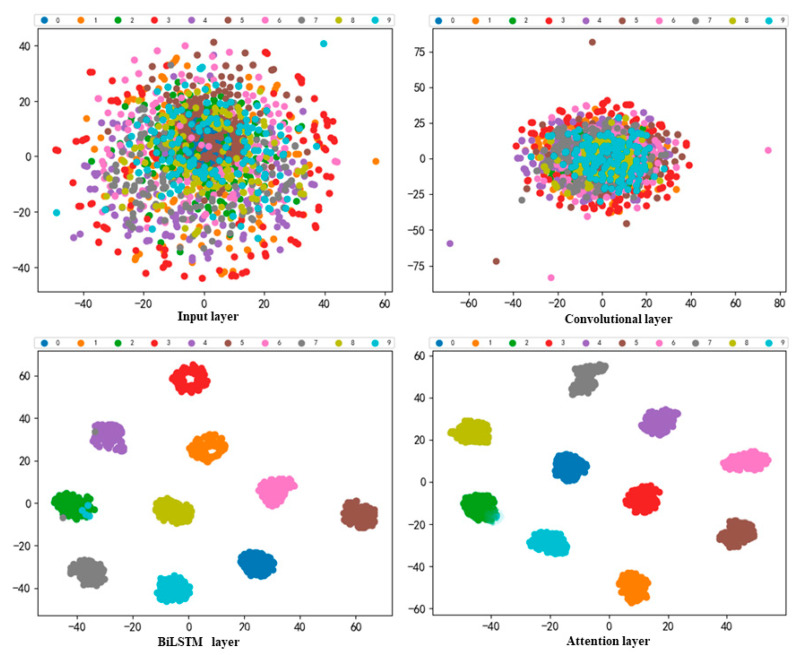
Visualization results of t-SNE for model feature extraction.

**Figure 14 sensors-22-08906-f014:**
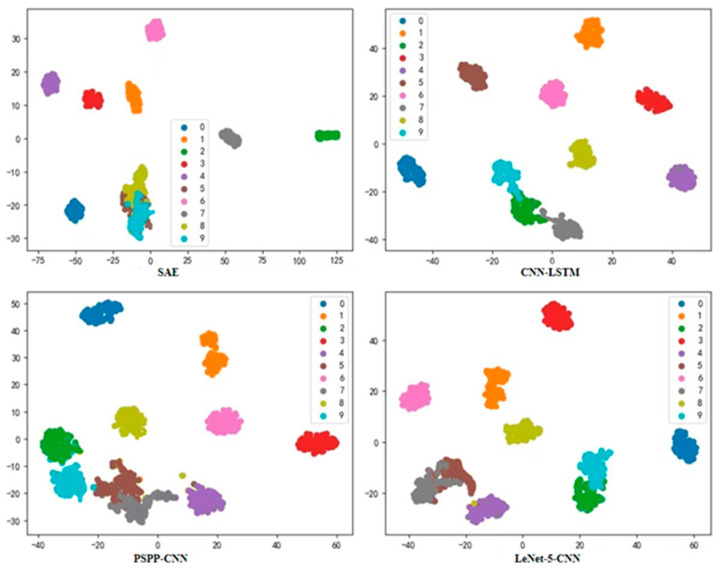
Visualization results of t-SNE in four model tests (for comparison with the proposed model).

**Figure 15 sensors-22-08906-f015:**
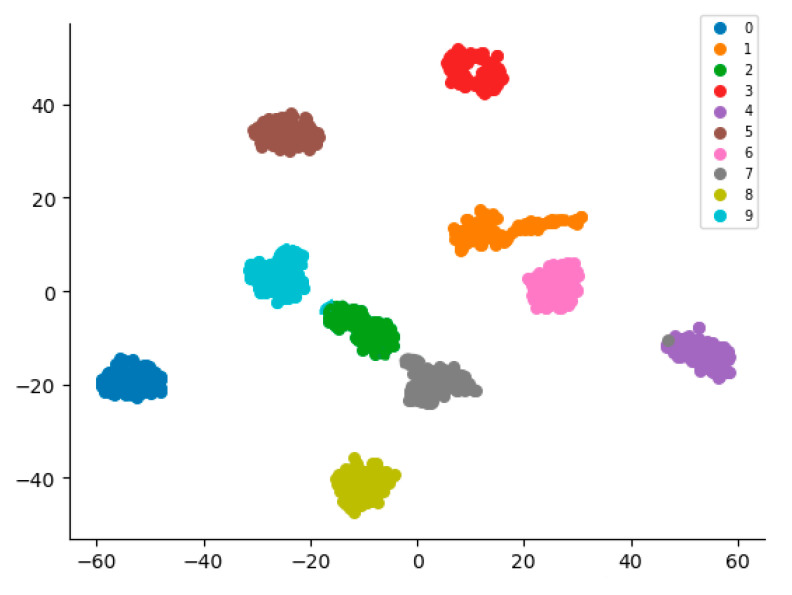
Visualization results of t-SNE in other data sets (from an industrial big data competition).

**Figure 16 sensors-22-08906-f016:**
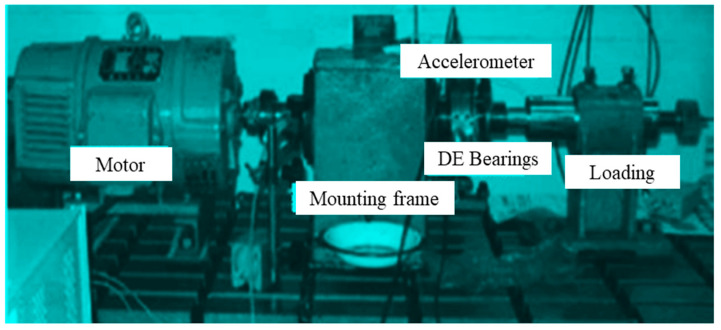
Practical rolling bearing fault data acquisition platform.

**Figure 17 sensors-22-08906-f017:**
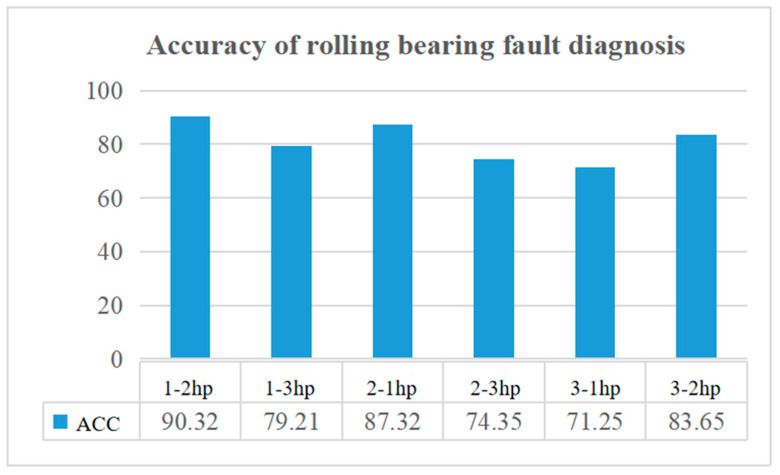
Fault diagnosis performance of rolling bearings under variable loads.

**Table 1 sensors-22-08906-t001:** CWRU bearing open data set sample processing.

Sampling Position	Bearing Type	Sampling Frequency	Load	Single-Point Loss Diameter
Drive end(DE) 6 o’clock	SKF6205	48 KHZ	0–2.205 N·m/s	0.007 mm, 0.014 mm, 0.021 mm

**Table 2 sensors-22-08906-t002:** Open CWRU bearing data set sample.

Sample Type	Sample Number	Sample Length	Training Sets	Test Set	Category Marker
Normal	1000	470	700	300	0
Outer ring fault (07 in)	1000	470	700	300	1
Outer ring fault (14 in)	1000	470	700	300	2
Outer ring fault (21 in)	1000	470	700	300	3
Inner ring fault (07 in)	1000	470	700	300	4
Inner ring fault (14 in)	1000	470	700	300	5
Inner ring fault (21 in)	1000	470	700	300	6
Rolling body fault (07 in)	1000	470	700	300	7
Rolling body fault (14 in)	1000	470	700	300	8
Rolling body fault (21 in)	1000	470	700	300	9

**Table 3 sensors-22-08906-t003:** Comparison of the accuracy of the proposed model with other models in rolling bearing fault diagnosis for CWRU.

Fault Diagnosis Model	Classifier	Percentage of Training Samples	Average Accuracy
ADCNN	Softmax	50%	97.9%
CNN	Softmax	90%	92.6%
LeNet-5-CNN	FC layer	83%	99.79%
IDS-CNN	Softmax	80%	98.92%
PSPP-CNN	Softmax	67%	99.19%
AOCNN	Softmax	50%	99.19%
SAE	ELM	50%	99.61
DBN	Softmax	N/A	98.8%
CNN-LSTM	Softmax	83%	99.6%
DC-GAN	SVM	96%	86.33%
GAAN-SDAE	Softmax	78%	99.2%
**The proposed**	**Softmax**	**70%**	**99.98%**

## Data Availability

Available Based on Request. The datasets generated and/or analyzed during the current study are not publicly available due to extension of the submitted research work, but are available from the corresponding author on reasonable request.

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
