# Peer review of "Rolling Bearing Fault Diagnosis Using Hybrid Neural Network with Principal Component Analysis"

_sensors, 2022, doi:10.3390/s22228906_

Round 1
Reviewer 1 Report
In this study, hybrid deep learning model is not new and many references had reported related results.
Author Response
- Thank you for your assessment of the overall innovation of our article, which has been very beneficial. Innovations are divided into original innovations, methodological improvements, and applied innovations. However, initial innovations are, after all, rare and do not take place step by step. More scientific work is done on the basis of previous work, gradually expanding and refining it, seeking methodological improvements and developments in application areas. Deep learning has also undergone several generations of continuous exploration and various attempts to become as powerful as it is today. We believe that this paper belongs to the application innovation of deep learning methods. For the real machine operation process, the fault generation and variable working condition are random, how to propose an intelligent fault diagnosis algorithm with qualified robustness and generalization performance is especially important, focusing on how to combine the existing deep learning framework for proper modification and test experiment design, so as to realize the fault diagnosis under the real working condition operation.
Reviewer 2 Report
I found your article very interesting titled “End-to-end hybrid neural network with principal component analysis for rolling bearing fault diagnosis”, but in my opinion below remarks would improve your manuscript under the scientific level.
Comments and Suggestions for Authors:
1. In the Abstract please specify the novelty of your research. What do you mean by the variable load?
2. I strongly disagree with the opinion that the bearing damage is very complex (Line 31). The linear methods such as FFT are giving precise information on the type of damage.
3. What I miss in the Introduction is the impact of radial internal clearance on the bearing’s dynamics and bearing’s life. I suggest to fulfil the list of references with following positions:
· Effect of radial clearance on ball bearing’s dynamics using a 2-DOF model. International Journal of Simulation Modelling, 2021, 20(3), pp. 513-524.
· Vibration characteristics of rotor-bearing system with andular misalignment and cage fracture: Simulation and experiment. Mechanical Systems and Signal Processing, 2023, 182, 109545.
4. Please explain what is the 1-2hp, 1-3hp… load? Is it a horsepower? I suggest it to write in standard units.
5. I don’t see the FFT analysis for the faults for studied bearing.
6. Why there is so strong difference between model in the percentage of training samples?
7. What are the components in the PCA?
8. What is the reference for Figure 13?
9. In my opinion, the accuracy of the model at 70% is too low, despite of the impact of high load.
10. Please specify the further steps of your research.
Author Response
Thank you for your overall comments on our article, we will do our best to improve our science in the process of revising the manuscript.
- In the abstract, we focus on our proposed model framework and point out our final experimental results. There mainly are two core innovations: firstly, it achieves higher accuracy than the existing models in terms of common data, secondly, it considers various random conditions when designing the test experiments, and performs robustness and generalization performance tests under the extreme conditions, and the test results are overall satisfactory. The overall results are satisfactory. It is worth noting that the so-called variable load is all the workload fluctuations that may occur during the actual machine operation.
- Thank you for your advice. As you said, the type of bearing damage can be identified by the linear FFT method, which is indeed simple and efficient. But the complexity highlighted in the article is the mechanism of bearing failure, that is, if the analysis and fault diagnosis is done from the point of view of mechanism modeling, it has some complexity, but it is also a very popular and in-depth research.
- Thank you for adding to our introduction section. The study of the effect of radial internal clearance on bearing dynamics and bearing life is a good reference for our study of the generation and diagnosis of bearing failures. It is because of your research that we have been able to think more about the problem of generating bearing failures, and we have added this description to the introduction, which we hope will meet with your approval.
- The 1-2hp, 1-3hp... load refers to the power load of the motor, i.e. horsepower, 1hp=0.735kw=0.735N.m/s. Your suggestion is very meaningful, and we try hard to standardize the unit of horsepower. But in some special places in the text, we use hp instead of N.m/s in order to express the meaning simply, hope to get your understanding and recognition.
- We are very sorry that the original manuscript did not have FFT analysis for the failure data. We know that FFT analysis is a simple and effective method to analyze bearing failure data by time domain, frequency domain, etc. It is a common method for data processing, so thank you very much for such a good comment. In the revised manuscript, we have performed an effective FFT analysis in time, and we hope our revision can meet your requirements.
- Thank you very much for your query and your comment is correct. Indeed, training an ML model on the same dataset with different proportions of training samples will produce different results. Therefore, it is not difficult to find that a model will have an optimal training sample ratio, and in the actual study results, we all would like to show the best model results, although good models may actually also perform well in a differentiated training sample. When we describe the diagnostic accuracy of a model, we are actually stating the maximum diagnostic accuracy of that classification model, so the proportion of training samples under the maximum accuracy correspondence is different for each model. It is easy to understand that when we compare the diagnostic accuracy of classification models, we have actually tried many training sample ratios, and what we are comparing is also the maximum diagnostic accuracy of a model.
- The original composition of PCA is actually the 470 dimensions of our original data set (CWRU bearing data set sample), and the dimensionality of the data after 30% dimensionality reduction is 330, and we use the final 330 dimensions of data to train the model, which can effectively reduce the computation and improve the performance of the model.
- Figure 13, or Figure 16 in the revised version, shows part of the equipment used in the study to do cross-platform testing, the equipment was used in actual production at the factory.
- In general, 70% fault diagnosis accuracy is a very poor performance indeed, so it is doubtful. We have reasonably considered all possible variable load cases in real life, and it can be seen from real test experiments that the model achieves a good performance over 83% for normal variable complexity (1-2hp or 2-1hp, 2-3hp or 3-2hp). 72.8% of the ultimate variable load diagnostic accuracy is a very good case for random conditions and is obtained from our real obtained from experimental tests.
- Thank you for your interest in our future research work, which is one of the greatest motivations for our efforts.. In the further steps of our research, we will design more practical test experiments and sufficiently overcome the randomness caused by model testing to conduct repeated cross-platform tests. Moreover, we will do our best to deploy and improve the intelligent fault diagnosis model proposed in this study as soon as possible.
Reviewer 3 Report
In this study, the authors proposed a deep-learning architecture for rolling bearing fault diagnosis. Although a promising performance has been achieved, some major points should be addressed as follows:
1. There must have external validation data to evaluate the performance of model on unseen data.
2. It is suggested to conduct cross-validation during the training process.
3. Uncertainties of model should be reported.
4. When comparing the performance among methods/models, the authors are suggested to provide some statistical tests to see significant differences.
5. Methodology is described using a lot of model definitions without detailed information related to model implementation i.e., hyperparameters, weights, etc.
6. More discussions should be provided to explain the results. The current version just listed some results without in-depth discussions.
7. Deep learning (i.e., CNN, LSTM) is well-known and has been used in previous studies such as PMID: 34915158. Thus, the authors are suggested to refer to more works in this description to attract a broader readership.
8. Overall, English language and presentation style should be improved significantly.
9. Quality of figures should be improved.
10. The authors should discuss the possibility of overfitting on the models.
11. There must have space before reference numbers.
Author Response
- Your proposal is worthy of our continued efforts, although CWRU is internationally recognized as the best dataset, and training and testing using CWRU dataset is beneficial to train a more realistic model. In order to show the superiority of the model, testing of other datasets is necessary. In the preliminary work, we only considered CWRU data and wanted to apply our intelligent diagnostic model to endpoint deployment as soon as possible. In the revised manuscript, we used the official dataset of the Industrial Big Data Competition for testing across the data.
- Thank you for your sincere suggestion. In our actual model training process, we have adopted the cross-validation method as a way to ensure the effectiveness of the model training, and it can well prevent the occurrence of overfitting during the model training process. In this paper we describe our specific cross-validation training method.
- Your advice is constructive and much appreciated. As you said, any deep learning machine learning model is based on statistical principles and randomness is inevitable. Also when conducting practical experimental tests, there is a certain amount of randomness in the test data we obtain, which is explained in the discussion and analysis of experimental results section later in the article. We hope that you will find our explanations acceptable and thank you again!
- Thank you for your detailed suggestions for a more scientific report on our articles. To verify the superiority of the proposed model, we compared the classification of four existing more advanced deep learning fault diagnosis models (SAE, CNN-LSTM, PSPP-CNN, and LeNet-5-CNN) and performed a statistical analysis of t-SNE.
- Thank you for your interest in important factors such as the specific hyperparameters and weights of our models. To provide a more scientific and understandable description of the methodology used in the study, in terms of the specific hyperparameters and weights involved, etc., we have summarily detailed the methodology in the hope that it will be better reflected.
- Your suggestions are very useful and thank you for this comment which will sublimate the value and significance of this study very well. In the last two ends of the last chapter of the article, we deeply summarize the findings of this paper and provide an in-depth analysis, and we hope to receive more guidance from you.
- Thank you very much for your sincere advice, we apologize for not referring to enough literature due to the limitations of our review of the relevant literature. When introducing deep learning methods, I am more than happy to go ahead and add the references you have recommended.
- Thank you for your suggestions on the English language and style of expression in our text, and all the language issues will be embellished at a later stage.
- Your advice is very important, we did not put much thought into improving the quality of the figures when we made them simply to express our ideas, and we have improved the quality of some of the images in the revised manuscript.
- Thank you very much for your suggestion. In the revised manuscript, we describe the possibility of overfitting of the model and propose a training method for 10-fold cross-validation, which is thus avoided.
- You are so careful that I have to go and thank you for your careful review of our articles. We hope that you will point out more details about our articles and we will try to improve them to meet your needs and those of our readers.
Round 2
Reviewer 2 Report
Dear Authors,
you have provided the answer to all remarks, which I can accept. I will recommend the manuscript for its publishing in its present form.
Reviewer
Reviewer 3 Report
My previous comments have been addressed well.